

# Benchmark levels for the consumptive water footprint of crop production for different environmental conditions: a case study for winter wheat in China

La Zhuo[1], Mesfin M. Mekonnen[1], Arjen Y. Hoekstra[1]

[1]Twente Water Centre, University of Twente, Enschede, 7500AE, The Netherlands

*Correspondence to*: L. Zhuo (l.zhuo@utwente.nl; zhuo.l@hotmail.com)





**Abstract.**
Meeting growing food demands while simultaneously shrinking the water footprint (WF) of agricultural production is one of
the greatest societal challenges. Benchmarks for the WF of crop production can serve as a reference and be helpful in setting
WF reduction targets. The consumptive WF of crops, the consumption of rainwater stored in the soil (green WF) and the
consumption of irrigation water (blue WF) over the crop growing period, varies spatially and temporally depending on
environmental factors like climate and soil. The study explores which environmental factors should be distinguished when
determining benchmark levels for the consumptive WF of crops. Hereto we determine benchmark levels for the consumptive
WF of winter wheat production in China for all separate years in the period 1961-2008, for rain-fed versus irrigated
croplands, for wet versus dry years, for warm versus cold years, for four different soil classes and for two different climate
zones. We simulate consumptive WFs of winter wheat production with the crop water productivity model AquaCrop at a 5
by 5 arc min resolution, accounting for water stress only. The results show that (i) benchmark levels determined for
individual years for the country as a whole remain within a range of ±20% around long-term mean levels over 1961-2008; (ii)
the WF benchmarks for irrigated winter wheat are 8-10% larger than those for rain-fed winter wheat; (iii) WF benchmarks
for wet years are 1-3% smaller than for dry years, (iv) WF benchmarks for warm years are 7-8% smaller than for cold years,
(v) WF benchmarks differ by about 10-12% across different soil texture classes; and (vi) WF benchmarks for the humid zone
are 26-31% smaller than for the arid zone, which has relatively higher reference evapotranspiration in general and lower
yields in rain-fed fields. We conclude that when determining benchmark levels for the consumptive WF of a crop, it is useful
to primarily distinguish between different climate zones. If actual consumptive WFs of winter wheat throughout China were
reduced to the benchmark levels set by the best 25% of Chinese winter wheat production (1224 $m^3 t^{-1}$ for arid areas and 841
$m^3 t^{-1}$ for humid areas), the water saving in an average year would be 53% of the current water consumption at winter wheat
fields in China. The majority of the yield increase and associated improvement in water productivity can be achieved in
southern China.



## 1 Introduction

Half of the large river basins in the world face severe blue water scarcity for at least one month a year (Hoekstra et al., 2012). Agriculture is the largest consumer of water in the world and therefore responsible for a large part of the water scarcity in the world. Still, global food demand continues to increase, due to growing populations and changing diets. Meeting growing food demands and simultaneously reducing the water footprint (WF) of agricultural production is therefore one of the greatest societal challenges of our time (Foley et al., 2011;Hoekstra and Wiedmann, 2014). Increasing water productivity (t $m^{-3}$) in croplands, i.e. reducing the WF of crops ($m^3$ $t^{-1}$), is recognized as an important way of producing more with the same water (or producing the same with less water). In order to know what is a reasonable production level given a certain amount of water consumption (or what is a reasonable volume of water consumption given a certain production level), we need reference values that indicate reasonable WF levels. (Hoekstra, 2014, 2013) has proposed to develop WF benchmarks for this purpose, which can be used for setting WF reduction targets. Such benchmarks could be global, but would preferably be context-specific, given the fact that the WF of growing a crop varies as a function of environmental factors such as climate and soil (Mekonnen and Hoekstra, 2011;Siebert and Doll, 2010;Tuninetti et al., 2015).

The WF of a crop is determined by both environmental conditions (e.g. climate, soil texture, $CO_2$ concentration in the air, groundwater level) and managerial factors (e.g. application of fertilizers and pesticides, irrigation technology and strategy, mulching practice) (Zwart et al., 2010;Mekonnen and Hoekstra, 2011;Brauman et al., 2013). Benchmarks for the WF of growing a crop can, for example, be set by looking at what WF level is not exceeded by the best 20-25% of the total production in an area. Alternatively, benchmarks can be determined by estimating the WF associated with best-available technology and management practice (Hoekstra, 2014, 2013). Mekonnen and Hoekstra (2014) followed the first approach, by establishing global benchmarks for both the consumptive (green plus blue) WF and the degradative (grey) WF for a large number of crops, based on estimated WF values for 1996-2005 at a spatial resolution of 5 by 5 arc minute. Chukalla et al. (2015) followed the second approach and explored reduction potentials of consumptive WFs for a few crops by applying different alternative irrigation techniques and strategies and different alternative mulching practices. They found that the highest reduction (~29%) in the consumptive WF of a crop could be achieved when applying drip/subsurface drip irrigation in combination with deficit irrigation and synthetic mulching.

Research in developing benchmark levels for the consumptive WF of crop production is still in its infancy. An important question that has been insufficiently addressed is which environmental factors should play a role when developing WF benchmarks. It is nice to have one global benchmark for the consumptive WF per crop, as a global reference, like the ones developed by Mekonnen and Hoekstra (2014), but it remains unclear whether it is reasonable to expect the same water productivity under different environmental conditions. In their global analysis, Mekonnen and Hoekstra (2014) found that a crop in a temperate climate generally has a smaller WF than the same crop in a tropical climate, but this can still be due to



other factors (e.g. better management practices in temperate climates), so that this is not a sufficient finding to diversify
benchmark levels based on the distinction between temperate and tropical. Besides, even though Mekonnen and Hoekstra
(2014) found a difference between different climates, for each crop considered it was found that the 10% best global
production (e.g. with smallest WFs) were always at least partly in the tropics as well. In other words, a WF benchmark
developed in the temperate part of the world still offers a reference value that can be achieved in the tropics as well. Next to
climate also soil affects evapotranspiration and yield and thus the WF of a crop. (Tolk and Howell, 2012), for example,
analyse the variation of consumptive WFs of sunflower in relation to different types of soils. There has not been yet, though,
a systematic study looking at how environmental factors influence the consumptive WFs of crops and to which extent it
makes sense to diversify WF benchmark levels based on specific environmental factors.
The current study aims to contribute to this discussion through an explorative study for winter wheat in China. We explore
which environmental factors should be distinguished when determining benchmark levels for the consumptive WF of crops.
We subsequently determine benchmark levels for the consumptive WF of winter wheat production in China for all separate
years in the period 1961-2008, for rain-fed versus irrigated croplands, for wet versus dry years, for warm versus cold years,
for four different soil classes and for two different climate zones. Winter wheat in China accounts for 95% of total wheat
production in China, which is the world biggest wheat producer (FAO, 2014). Winter wheat covers 96% of China's
harvested wheat area and occurs across China's different climate zones (NBSC, 2013). In order to avoid interference from
managerial factors that cause differences in evapotranspiration and yield, we simulate WFs by means of the water
productivity model AquaCrop, at a resolution of 5 by 5 arc minute, considering only water stress and not taking into account
other stresses such as from soil fertility, salinity, frost, or pest and diseases.
**2 Method and data**
The consumptive WF of growing a crop ($m^3 t^{-1}$) equals the total actual evapotranspiration (ET, $m^3 ha^{-1}$) over the cropping
period divided by the crop yield (t $ha^{-1}$). The ET and crop yield were simulated at daily basis, at 5 by 5 arc min resolution,
with FAO's crop water productivity model AquaCrop (Hsiao et al., 2009;Raes et al., 2009;Steduto et al., 2009), run for the
whole period 1961-2008. Compared to other crop growth models, AquaCrop has a significantly smaller number of
parameters and better balances between simplicity, accuracy and robustness (Steduto et al., 2007). AquaCrop has been
applied in WF accounting at both field (Chukalla et al., 2015) and river basin level at high spatial resolution (Zhuo et al.,
2016). Data on monthly precipitation, reference evapotranspiration ($ET_0$) and temperature at 30 arc min resolution were
taken from the CRU-TS 3.10 dataset (Harris et al., 2014). Soil texture data were obtained from Dijkshoorn et al. (2008). For
hydraulic characteristics for each type of soil, the indicative values provided by AquaCrop were used. Data on total soil
water capacity were obtained from Batjes (2012). In order to avoid the effects of non-environmental factors (e.g. technology,
fertilization) on crop growth, only water stress is considered, which is determined by the water availability in the root zone.





For irrigated fields, we assume that the applied irrigation volumes are equal to the net irrigation requirement. We simulated
winter wheat production per grid cell over the years based on the irrigated and rain-fed harvested areas of around the year
2000, as obtained from Portmann et al. (2010) (Fig. 1) in order to avoid in the simulations the effects of changes in where
how much wheat is grown.
Following Mekonnen and Hoekstra (2014), benchmark levels for the consumptive WF of crop production were determined
by ranking the grid-level WF values from the smallest to the largest against the corresponding cumulative percentage of total
crop production.
In order to analyse differences in consumptive WFs in relatively dry versus relatively wet years, we evenly group the forty-
eight considered years (1961-2008) into relative dry, average and relatively wet years. We ranked the years based on the
annual precipitation over the cropping area of winter wheat in China (Fig. 2a) and classified the sixteen years with the lowest
precipitation into the group of dry years and the sixteen years with the highest precipitation into the group of wet years, with
the other sixteen years remaining for the group of average years. The average annual precipitation levels of the relatively dry,
average and relatively wet years are 760, 799 and 850 mm $y^{-1}$, respectively.
We also grouped the years considered into relatively cold, average and relatively warm years based on annual mean
temperature (Fig. 2b) and into years with relatively low, average and high $ET_0$ (Fig. 2c). The average annual mean
temperatures of the relative cold, average and warm years are 10.7, 11.2 and 11.8 ℃, respectively. The average annual $ET_0$
values in the three categories of years are 874, 896 and 927 mm $y^{-1}$.
For determining WF benchmarks for different soil texture classes, the soil types in the USDA (U.S. Department of
Agriculture) soil texture triangles were grouped into four soil classes (Raes et al., 2011): sandy soils, loamy soils, sandy
clayey soils, and silty clayey soils. Each soil class has different ranges of field capacity, permanent wilting point and
saturated water content (Table 1). The difference between soil water content and permanent wilting point defines the total
available soil water content in the root zone. Given certain soil water content, a soil with a higher field capacity has less deep
percolation. With the same water input from precipitation or irrigation and the same  soil water content, soils with a smaller
saturated soil water content will generate more surface runoff (Raes et al., 2011). Figure 3 shows the spatial distribution of
the four soil classes across mainland China.

For determining WF benchmarks for different climate zones, we classify climate based on UNEP's aridity index (AI)
(Middleton and Thomas, 1997, 1992). The AI is an indicator of dryness, defined as the ratio of precipitation to reference
evapotranspiration, with five levels of aridity: hyper-arid (AI< 0.05), arid (0.05 < AI < 0.2), semi-arid (0.2 < AI < 0.5), dry
sub humid (0.5 < AI < 0.65), and humid (AI > 0.65). To determine the geographic spread of the five climate zones in China



we used the data on annual precipitation and $ET_0$ averaged over the period 1961-2008 at 30 by 30 arc min resolution (Harris
et al., 2014) (Fig. 4). In the current study, we group the five climate zones into two broad zones: the arid-semi-arid (Arid)
zone (AI < 0.5) and the humid-semi-humid (Humid) zone (AI >0.5).

## 3 Result

### 3.1 Benchmark levels for the consumptive WF as determined for different years and for rain-fed and irrigated croplands separately

We calculated the benchmark levels at different production percentiles for the consumptive WF of winter wheat ($m^3 t^{-1}$) for
the country as a whole, year by year, for the period 1961-2008. The results are summarized in Fig. 5. The benchmarks,
determined per year and per production percentile, generally vary within ±20% of the long-term mean value over the period
1961-2008. We find that the best 10% of winter wheat production in China (with smallest WFs) has a maximum long-term
average consumptive WF of 777 $m^3 t^{-1}$, which is larger than the maximum consumptive WF of the best 10% of wheat
production globally (592 $m^3 t^{-1}$) that was reported by Mekonnen and Hoekstra (2014). We note here that the figures are not
fully comparable, because Mekonnen and Hoekstra (2014) consider total wheat (both spring and winter wheat), use another
model and consider another period. We find that the best 20% of winter wheat production in China has a maximum long-
term average consumptive WF of 825 $m^3 t^{-1}$, which is *smaller* than the reported maximum consumptive WF of the best 20%
of wheat production globally (992 $m^3 t^{-1}$). Finally, we find that the best 25% of winter wheat production in China has a
maximum long-term average consumptive WF of 849 $m^3 t^{-1}$, which is again *smaller* than the maximum consumptive WF of
the best 25% of wheat production globally (1069 $m^3 t^{-1}$).
The national average consumptive WF of rain-fed winter wheat (1120 $m^3 t^{-1}$) is larger than the national average consumptive
WF of irrigated winter wheat (1075 $m^3 t^{-1}$). However, the benchmark levels determined by the best 10%, 20% and 25% of
production for rain-fed winter wheat are lower than for irrigated winter wheat. The reason is that the yields in rain-fed
production are generally higher than the yields in irrigated production at the same benchmark percentile. The highest rain-fed
yields occur in the southern wet area with sufficient precipitation over the cropping period, so that little water stress results
in high rain-fed yields. The WF benchmarks for irrigated winter wheat are 8% (for the 10[th] production percentile) to 10%
(for the 25[th] production percentile) higher than for rain-fed winter wheat.

### 3.2 Benchmark levels for the consumptive WF for dry versus wet years

In a relatively dry or wet year, when considering winter wheat areas in China as a whole, we do not find typically different
consumptive WFs in winter wheat production (Table 2). The WF benchmarks are consistently higher in dry than in wet years
(1-3%), but the differences between benchmark levels for the consumptive WF for dry versus wet years are small compared
to the variations within the dry and wet year categories (±11-14%).



## 3.3 Benchmark levels for the consumptive WF for warm versus cold years

Overall, considering irrigated and rain-fed croplands together, WF benchmarks for relatively warm years are 7-8% smaller than for relatively cold years, which is not much when seen in the context of fluctuations in the WFs within the three temperature categories (Table 3). In irrigated areas, WF benchmarks for warm years are 11% smaller, on average, than for cold years. In rain-fed areas, WF benchmarks for warm years are smaller than for cold years as well, but WF benchmarks in average years are not in between the WF benchmarks found for cold and warm years but higher than both. The lower values in cold years relate to lower ET, while the lower values in warm years relate to higher yields.

The findings when considering different $ET_0$ classes are similar when looking at the different temperature classes (Table 4). Overall, considering irrigated and rain-fed croplands together, WF benchmarks for years with high $ET_0$ are on average 5% smaller than for years with average $ET_0$ and only 2% smaller than for years with low $ET_0$. Again, differences between consumptive WFs for years with relatively low or high $ET_0$ are small when seen in the context of fluctuations in the WFs within the three $ET_0$ categories ($\pm$3-6%).

## 3.4 Benchmark levels for the consumptive WF for different soil classes

Tables 5 shows the consumptive WFs of winter wheat at different production percentiles in four soil classes in China. The simulated winter wheat production in sandy clayey soils accounts for 60% of national total, followed by the production in sandy soils (24%), silty clayey soils (8%) and loamy soils (8%) in average over the studied period. No consistent trends can be observed when we compare the benchmarks across the different soil classes. Overall, when we take irrigated and rain-fed fields together, the WF benchmarks for sandy soils are 10-12% lower than the WF benchmarks for loamy soils. More specifically, we find that the WF benchmarks for irrigated winter wheat in sandy soils are about 15% smaller than the WF benchmarks for the other three soil classes, due to relatively low ET. Without water stress, as is the case in the irrigated croplands, soil evaporation from sandy soils is less than from the other soil types because of the fast percolation of water below the root zone in the sandy soils, causing lower ET over the cropping period (Asseng et al., 2001). At rain-fed fields with limited water availability, crop yields are mainly affected by the soil water holding capacity. Therefore, consumptive WFs in sandy soils are larger than in the other three soils, due to the smaller crop yield in case of poorer water holding capacity. The observed differences in WFs of winter wheat in different soil classes agree with the experimental observations by Tolk and Howell (2012) for the case of irrigated sunflower in a semiarid environment as well as with the fieldwork-based simulations by Asseng et al. (2001) for irrigated and rain-fed wheat in the Mediterranean climatic region of Western Australia.



**3.5 Benchmark levels for the consumptive WF for different climate zones**
Consumptive WFs of winter wheat at different production percentiles in arid and humid zones in China are shown in Table 6.
Significant differences between the benchmarks for different climate zones can be observed. Overall, considering irrigated
and rain-fed croplands together, WF benchmarks for the humid zone are 26% (for the 10th production percentile) to 31%
(for the 25th production percentile) smaller than for the arid zone. The WF benchmarks for winter wheat in China as a whole
(when we take the arid and humid zones together) are close to the benchmarks for the humid zone, caused by the fact that
most (96% in average over the study period) of the simulated winter wheat production in China occurs in the humid zone.
In the irrigated areas, WF benchmarks for the humid zone are 26-30% smaller than for the arid zone; in the rain-fed areas,
they are 29-43% smaller. The relatively large WFs in rain-fed fields in the arid zone logically follow from the water stress
and resultant low yields. For the irrigated fields, the larger WFs in the arid zone are caused by the relatively high $ET_0$ and ET.
The results confirm the findings from previous studies that the WF of crops, especially rain-fed crops, is negatively
correlated with precipitation and positively correlated with $ET_0$ (Zwart et al., 2010;Zhuo et al., 2014).  The differences
between the WF benchmarks for irrigated and rain-fed winter wheat are 7-9% in the humid zone and 3-11% in the arid zone.
Figure 6 shows, for both the humid and arid part of China, for the various winter wheat production areas whether they
contribute to the best 10% of national winter wheat production in that climate zone (in the sense of having smallest WFs), to
the next best 10%, to the best 5% after that, or to the worst 75% (with WFs beyond the $25^{th}$ percentile benchmark). Within
the arid zone, consumptive WFs below the $25^{th}$ percentile benchmark level were mostly located in Xinjiang province, with
relatively high irrigation density (~98% of the harvested area). In the humid zone, consumptive WFs below the $25^{th}$
percentile benchmark level were gathered in the southwest, where $ET_0$ is smaller than in other places (Fig. 4b).
**3.6 Water saving potential by reducing WFs to selected benchmark levels**
The WF benchmarks for different climate zones differ much more significantly (26-31%) than for different soils (10-12%).
WF benchmarks differ even less if we compare irrigated versus rain-fed fields (8-10%), warm versus cold years (7-8%), or
wet versus dry years (1-3%). Therefore, when determining benchmark levels for the consumptive WF of a crop, it seems
most useful to primarily distinguish between different climate zones, at least in the case of winter wheat in China. In this
section, we analyse the potential water saving if actual consumptive WFs of winter wheat throughout China were reduced to
the climate-specific benchmark levels set by the best 10% of Chinese winter wheat production (1042 $m^3 t^{-1}$ for arid areas and
776 $m^3 t^{-1}$ for humid areas), the best 20% of Chinese winter wheat production (1170 $m^3 t^{-1}$ for arid areas and 819 $m^3 t^{-1}$ for
humid areas), or the best 25% of Chinese winter wheat production (1224 $m^3 t^{-1}$ for arid areas and 841 $m^3 t^{-1}$ for humid areas).
Taking the estimated actual consumptive WFs of winter wheat in 2005, an average climatic year, as calibrated by the
provincial statistics on yield of winter wheat (NBSC, 2013), we find that consumptive WFs in 75% of the planted grids in





arid zones and in 96% of the planted grids in humid zones are over the 25th percentile benchmarks. This is largely due to
low actual versus potential yields. Figure 7 shows differences between actual provincial yields of winter wheat and the
simulated yield potentials from the current study (assuming no crops stresses except water stress in rain-fed areas). The
largest yield gaps occur in the southern provinces in the humid zone. The largest yield gap was observed in Fujian province.
Table 7 shows the (green plus blue) water saving that would be achieved if actual consumptive WFs of winter wheat
everywhere in China were reduced to the climate-differentiated WF benchmark levels set by the 10th, 20th and 25th
percentiles of production, in an average year (2005). We find that if in both the arid and humid zone the actual consumptive
WFs were reduced to the respective $25^{th}$ percentile benchmark level, the water saving in an average year would be 53% of
the current water consumption at winter wheat fields in China, which is 201 billion $m^3 y^{-1}$ in absolute terms. We further find
that the water saving potential in the arid zone is substantially higher than in the humid zone.

## 3.7 Discussion

The consumptive WF of a crop in $m^3 t^{-1}$ most strongly depends on the crop yield in t $ha^{-1}$ and much less on the
evapotranspiration from the crop over the growing period in $m^3 ha^{-1}$ (Tuninetti et al., 2015;Mekonnen and Hoekstra, 2011).
For evaluating our simulations of crop growth, we compared the current simulated averaged yields of winter wheat of
Chinese provinces for 1961-1990 to the corresponding agro-climatic attainable yields at different agricultural input levels in
the GAEZ database (FAO/IIASA, 2011) (Fig. 8). The GAEZ agro-climatic attainable yields account for different levels of
yield constraints from four factors in addition to water stress: (i) pest, diseases and weed damages on plant growth, (ii) direct
and indirect climatic damages on quality of produce, (iii) efficiency of farming operations, and (iv) frost hazards. Current
simulated yields of irrigated winter wheat are closest to the agro-climatically attainable yields with intermediate input levels
and the yields of rain-fed winter wheat are closest to the agro-climatically attainable yields with high input levels. The
simulated national average yield in the current study (6.5 t $ha^{-1}$) is 23% higher than the attainable wheat yield for China in
the year 2000 (5.3 t $ha^{-1}$) estimated by Mueller et al. (2012).
Further research could explore whether crop varieties used should play a role when developing WF benchmarks, given the
fact that some crop varieties may inherently be more productive than others. On the other hand, one could also consider that
choosing a productive crop variety is part of the managerial choices. Since crop variety is not a given environmental
condition but a choice, one could argue that accepting a less strict WF reference level for a less productive crop variety
cannot be justified.

An important remaining research question is also how combinations of specific techniques and practices can actually lead to
the WF reductions that will be necessary in different locations if Chinese government would adopt certain WF benchmarks
as targets to achieve greater water productivity. Suppose, for example, that two WF benchmarks for winter wheat were




adopted in China: 1224 m$^3$ t$^{-1}$ for arid areas and 841 m$^3$ t$^{-1}$ for humid areas. Although the simulations suggest that these levels
are feasible throughout the arid and humid zone, respectively, whatever is the soil, whether fields are rain-fed or irrigated,
whether it is a cold or warm year, and whether it is a dry or wet year, in some places it will be harder and more would need
to be done than in other places.
We studied benchmarks for combined green and blue WFs and did not look at each colour separately. For rain-fed lands, the
benchmark levels presented in this study are obviously green WF benchmarks. For irrigated lands, the presented benchmark
levels for overall consumptive WFs would need further specification into green and blue. Further research would need to be
done to translate a certain benchmark level for the overall consumptive WF of a crop into a specific blue WF benchmark
level per specific location as a function of the amount of rain per location, recognizing that the blue ratio in the WF will need
to be larger if less green water is available.

## 4 Conclusions

Based on the case of winter wheat in China we find that (i) benchmark levels for the consumptive WF determined for
individual years for the country as a whole remain within a range of ±20% around long-term mean levels over 1961-2008; (ii)
the WF benchmarks for irrigated winter wheat are 8-10% larger than those for rain-fed winter wheat; (iii) WF benchmarks
for wet years are on average 1-3% smaller than for dry years, (iv) WF benchmarks for warm years are on average 7-8%
smaller than for cold years, (v) WF benchmarks differ by about 10-12% across different soil texture classes; and (vi) WF
benchmarks for the humid zone are 26-31% smaller than for the arid zone, which has relatively higher ET$_0$ in general and
lower yields in rain-fed fields. Therefore, we conclude that when determining benchmark levels for the consumptive WF of a
crop, it is useful to primarily distinguish between different climate zones. We estimated that when in both the arid and humid
zone the actual consumptive WFs are reduced to climate-specific benchmark levels set by the 25$^{th}$ percentile of production,
the water saving in an average year would be 53% of the current water consumption at winter wheat fields in China, with
greatest relative savings in the arid zone.

## Author contribution

A. Y., L. Z. and M. M. M. designed the study. L. Z. carried it out. L. Z. prepared the manuscript with contributions from all
co-authors.



1 **Acknowledgement**

2 The work was partially developed within the framework of the Panta Rhei Research Initiative of the International

3 Association of Hydrological Sciences (IAHS).





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

**Tables**

Table 1. Soil classes.

| Soil class | Soil types | Soil water content (vol %) | | |
|---|---|---|---|---|
| | | Field capacity | Permanent wilting point | Saturation |
| Sandy | Sand, loamy sand, sandy loam | 9 - 28 | 4 - 15 | 32 - 51 |
| Loamy | Loam, silt loam, silt | 23 - 42 | 6 - 20 | 42 - 55 |
| Sandy clayey | Sandy clay, sandy clay loam, clay loam | 25 - 45 | 16 - 34 | 40 - 53 |
| Silty clayey | Silty clay loam, silty clay, clay | 40 - 58 | 20 - 42 | 49 - 58 |

Source: Raes et al. (2011).





Table 2. Benchmark levels for the consumptive water footprint (WF) benchmarks ($m^3 t^{-1}$) of winter wheat for relative dry, average and wet years in China.

| Crop | | Consumptive WF ($m^3 t^{-1}$) at different production percentiles* | | | |
| --- | --- | --- | --- | --- | --- |
| | | 10th | 20th | 25th | Average |
| Winter wheat | Dry years | 787±69 | 837±70 | 858±71 | 1103±82 |
| | Average years | 763±107 | 826±72 | 849±74 | 1073±97 |
| | Wet years | 770±68 | 813±60 | 838±50 | 1048±77 |
| Irrigated winter wheat | Dry years | 822±118 | 862±110 | 876±112 | 1095±110 |
| | Average years | 814±97 | 856±97 | 881±98 | 1078±93 |
| | Wet years | 799±97 | 850±100 | 870±96 | 1052±96 |
| Rain-fed winter wheat | Dry years | 757±44 | 802±57 | 812±56 | 1121±97 |
| | Average years | 736±62 | 771±70 | 783±70 | 1074±133 |
| | Wet years | 755±96 | 784±103 | 794±104 | 1164±561 |

* Data are mean ± S.D. for the years 1961-2008.





Table 3. National consumptive water footprint (WF) benchmarks ($m^3\ t^{-1}$) of winter wheat for relative cold, warm and average years in China.

| Crop | | \multicolumn{4}{c}{Consumptive WF ($m^3\ t^{-1}$) at different production percentiles*} |
| | | 10th | 20th | 25th | Average |
|---|---|---|---|---|---|
| Winter wheat | Cold years | 795 ±101 | 848 ±63 | 870 ±67 | 1103 ±96 |
| | Average years | 794 ±79 | 840 ±66 | 864 ±58 | 1087 ±82 |
| | Warm years | 732 ±42 | 788 ±58 | 811 ±57 | 1033 ±70 |
| Irrigated winter wheat | Cold years | 862 ±86 | 902 ±87 | 924 ±87 | 1121 ±86 |
| | Average years | 810 ±107 | 863 ±102 | 878 ±96 | 1083 ±93 |
| | Warm years | 763 ±96 | 804 ±93 | 824 ±96 | 1022 ±98 |
| Rain-fed winter wheat | Cold years | 760 ±59 | 791 ±68 | 798 ±69 | 1088 ±144 |
| | Average years | 772 ±95 | 821 ±99 | 831 ±100 | 1218 ±553 |
| | Warm years | 716 ±31 | 744 ±40 | 761 ±44 | 1053 ±63 |

* Data are mean ±S.D. for the years 1961-2008.





Table 4. National consumptive water footprint (WF) benchmarks ($m^3 t^{-1}$) of winter wheat for relative low-, high- and average-$ET_0$ years in China.

| Crop | | Consumptive WF ($m^3 t^{-1}$) at different production percentiles* | | | |
|---|---|---|---|---|---|
| | | 10th | 20th | 25th | Average |
| Winter wheat | Low-$ET_0$ years | 774 ±99 | 822 ±64 | 841 ±62 | 1065 ±82 |
| | Average years | 806 ±80 | 846 ±73 | 866 ±76 | 1095 ±107 |
| | High-$ET_0$ years | 741 ±51 | 808 ±62 | 839 ±58 | 1065 ±70 |
| Irrigated winter wheat | Low-$ET_0$ years | 831 ±111 | 874 ±108 | 892 ±106 | 1089 ±98 |
| | Average years | 820 ±105 | 868 ±96 | 887 ±96 | 1073 ±103 |
| | High-$ET_0$ years | 784 ±93 | 827 ±97 | 847 ±97 | 1064 ±102 |
| Rain-fed winter wheat | Low-$ET_0$ years | 749 ±55 | 774 ±56 | 781 ±54 | 1038 ±100 |
| | Average years | 784 ±90 | 828 ±98 | 841 ±98 | 1249 ±550 |
| | High-$ET_0$ years | 716 ±72 | 755 ±59 | 767 ±58 | 1072 ±78 |

* Data are mean ± S.D. for the years 1961-2008.





Table 5. Benchmark levels for the consumptive water footprint (WF)  ($m^3 t^{-1}$) of winter wheat for different soil classes in China.

| Crop | Soil class | Consumptive WF ($m^3 t^{-1}$) at different production percentiles* | | | |
|---|---|---|---|---|---|
| | | 10th | 20th | 25th | Average |
| Winter wheat | Sandy | 748±143 | 814±115 | 834±116 | 1017±125 |
| | Loamy | 846±53 | 912±77 | 928±73 | 1108±74 |
| | Sandy clayey | 788±76 | 848±61 | 881±66 | 1071±48 |
| | Silty clayey | 822±48 | 895±43 | 912±46 | 963±22 |
| Irrigated winter wheat | Sandy | 767±158 | 782±177 | 846±128 | 1000±126 |
| | Loamy | 931±91 | 937±93 | 996±70 | 1189±107 |
| | Sandy clayey | 879±98 | 932±98 | 969±102 | 1164±100 |
| | Silty clayey | 920±68 | 942±72 | 958±66 | 1070±52 |
| Rain-fed winter wheat | Sandy | 785±58 | 834±88 | 850±96 | 1151±272 |
| | Loamy | 757±77 | 822±73 | 843±73 | 1040±160 |
| | Sandy clayey | 764±66 | 799±68 | 818±70 | 1096±129 |
| | Silty clayey | 769±62 | 814±60 | 837±60 | 931±103 |

* Data are mean ±S.D. for the years 1961-2008.





Table 6. Benchmarks for the consumptive water footprint (WF) $(m^3\,t^{-1})$ of winter wheat for different climate zones in China.

| Crop | Climate zones | Consumptive WF $(m^3\,t^{-1})$ at different production percentile* | | | |
|---|---|---|---|---|---|
| | | 10th | 20th | 25th | Average |
| Winter wheat | Arid | 1042±100 | 1170±130 | 1224±125 | 1757±200 |
| | Humid | 776±70 | 819±66 | 841±66 | 1044±83 |
| | Overall | 777±72 | 825±67 | 849±65 | 1075±87 |
| Irrigated winter wheat | Arid | 1088±66 | 1205±73 | 1245±84 | 1399±163 |
| | Humid | 807±104 | 853±100 | 872±99 | 1055±97 |
| | Overall | 812±103 | 856±100 | 875±100 | 1075±99 |
| Rain-fed winter wheat | Arid | 1058±310 | 1311±406 | 1399±415 | 2919±1004 |
| | Humid | 749±70 | 784±78 | 795±79 | 1076±338 |
| | Overall | 750±70 | 785±78 | 796±78 | 1120±332 |

* Data are mean ±S.D. for the years 1961-2008.



Table 7. Water saving if actual consumptive water footprint (WF) of winter wheat everywhere in China were reduced to the climate-differentiated WF benchmark levels set by the 10th, 20th and 25th percentiles of production, in an average year (2005).

| Climate zones | Water saving when actual consumptive WF of winter wheat everywhere in China were to be reduced to a certain percentile benchmark level | | |
|---|---|---|---|
| | 10th | 20th | 25th |
| Arid | 83% | 81% | 80% |
| Humid | 49% | 46% | 45% |
| Overall | 56% | 54% | 53% |

* Data are mean ±S.D. for the years 1961-2008.





1  **Figures**

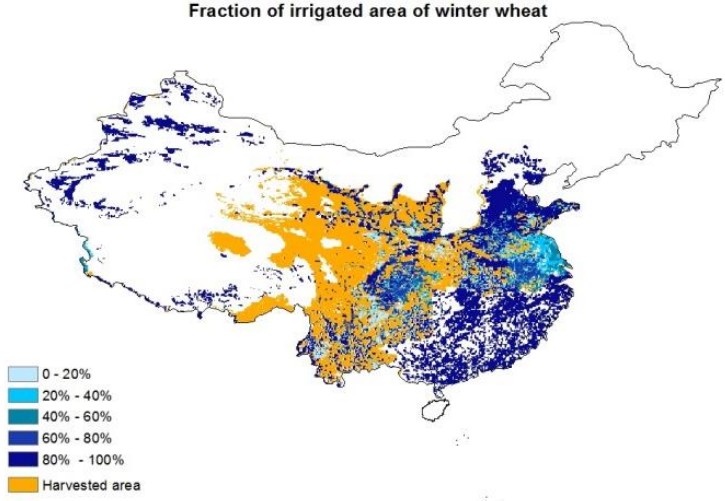

5  Figure 1. Harvested winter wheat areas in China in the year 2000 and fractions of the harvested areas irrigated. Data source:

6  Portmann et al. (2010).





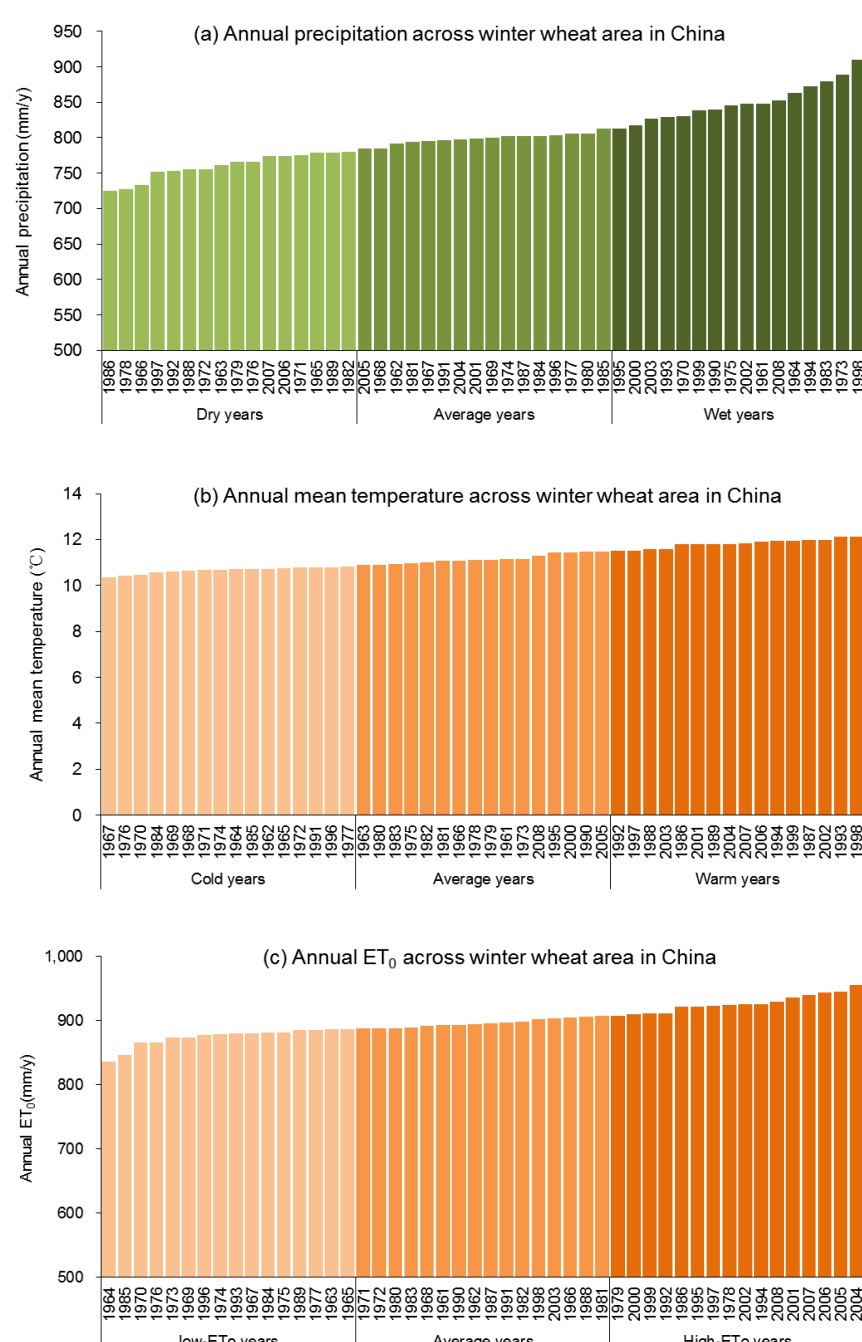

Figure 2. Annual precipitation (a), mean temperature (b), and $ET_0$ (c) over the cropping area of winter wheat in China for the years in the period 1961-2008, ranked from lowest to highest values. Data source: Harris et al. (2014).





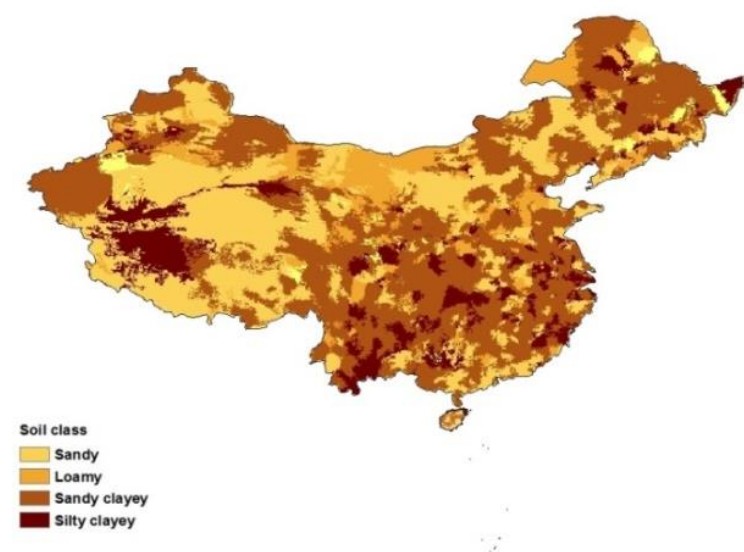

2 Figure 3. Soil classes across mainland China, generated from the ISRIC Soil and Terrain database for China. Data source:

3 Dijkshoorn et al. (2008).


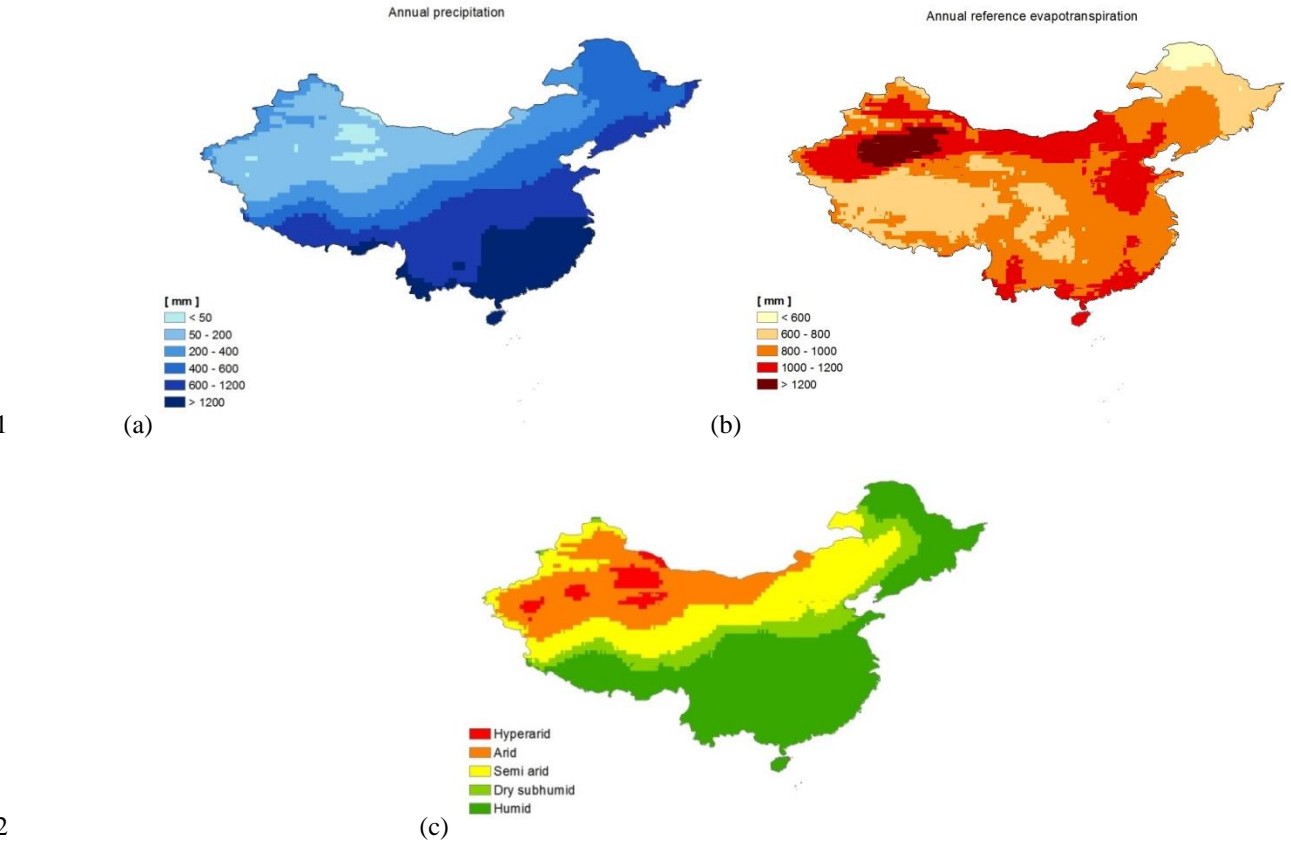

(a)      (b)
(c)
Figure 4. Zoning of annual precipitation (a), annual reference evapotranspiration (b), and aridity (c) in China (1961-2008).
Data source: Harris et al. (2014).





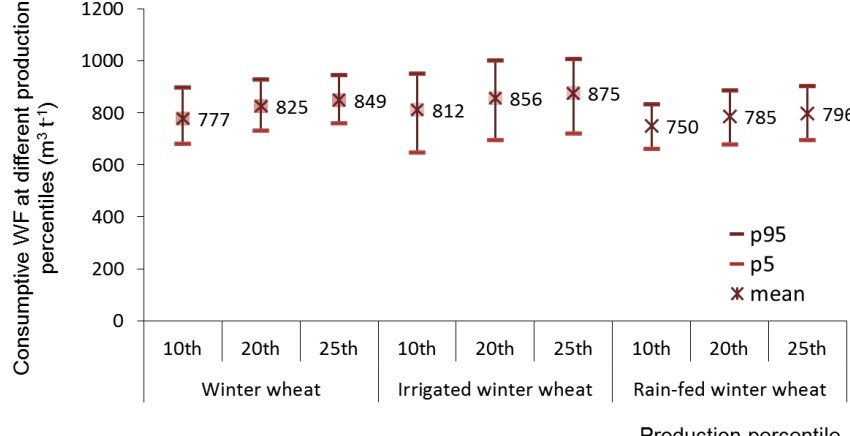

2 Figure 5. Benchmark levels for the consumptive water footprint (WF) of winter wheat in China at different production

3 percentiles, considering all separate years in the period 1961-2008. Cross marks refer to the mean values; ranges refer to the

4 5% - 95% of accumulative frequencies.





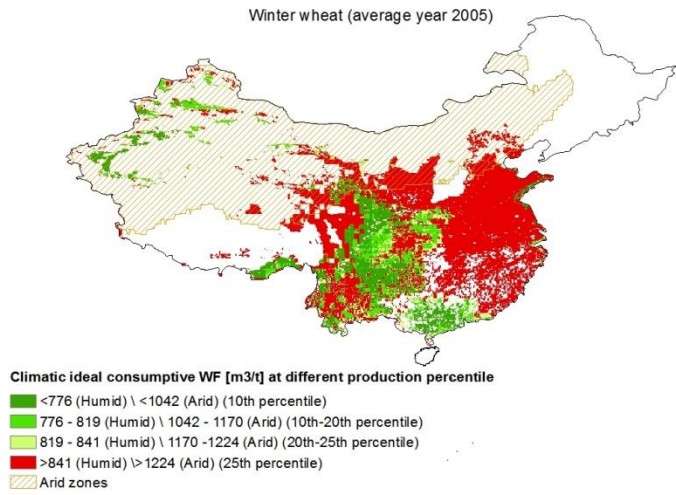

2 Figure 6. Simulated consumptive water footprints (WFs) of winter wheat, categorized into four classes (the best 10% of

3 production, the next best 10%, the second next best 5% and the worst 75% of production), accounting for different

4 benchmark levels for humid versus arid part of China, for the year 2005 (climatic average year).





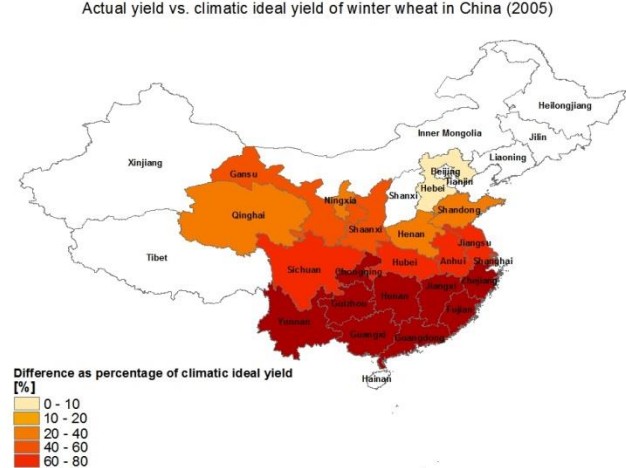

2 Figure 7. Differences between actual provincial yields of winter wheat in China in 2005 (NBSC, 2013) and simulated yields

3 from the current study (assuming no crop stress except for water stress in rain-fed areas), expressed as percentage of the

4 simulated yield.





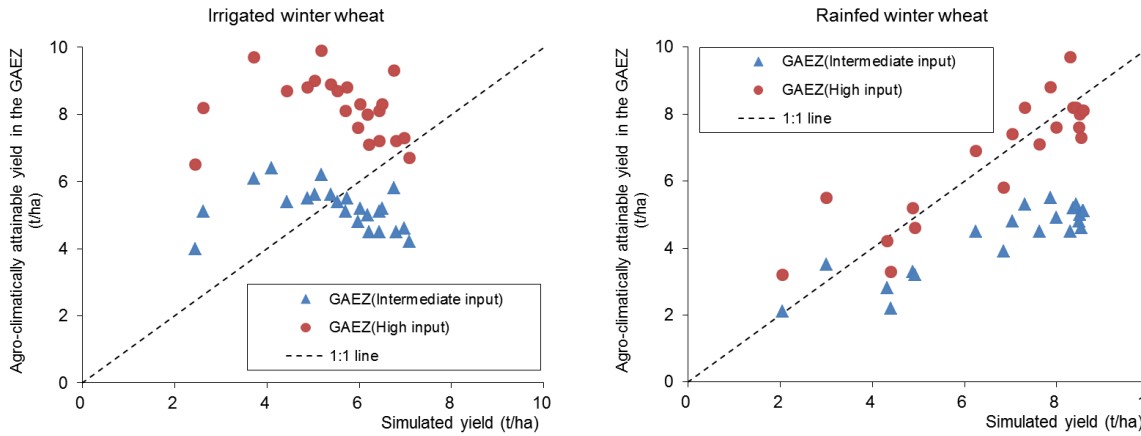

2 Figure 8. Comparison between the simulated yield of winter wheat and the agro-climatically attainable yield according to

3 (FAO/IIASA, 2011) at provincial level in China. Averaged over the period 1961-1990.

