# Peer review of "Benchmark levels for the consumptive water footprint of crop production for different environmental conditions: a case study for winter wheat in China"

_Hydrology and Earth System Sciences, 2016_

## Referee Comment (RC1) · Anonymous Referee #1 · 28 Apr 2016

The study presents an attempt to estimate benchmark levels for the consumptive water footprint of winter wheat in China by applying the crop model AquaCrop. Water footprints simulated for the period 1961-2008 are compared between dry and wet years, between warm and cold years, across soil types and between irrigated and rainfed wheat. Such an analysis is interesting in general and fits also well to the scope of the journal. However, I think that the manuscript requires substantial improvement before it may be considered for publication in HESS. May main points of criticism are:

1.) I completely miss a discussion on the relevance of the water footprints obtained in the present study. Why is this information needed? Farmers, for example, are not interested in optimizing water footprints; they are more interested in optimizing

their economic return. The attempt to optimize the water footprint to benchmark levels provided in this study may also be misleading from an environmental or ecological perspective because these water footprints can only be achieved when nutrients are not limiting, thus require high nutrient inputs and consequently high nutrient losses. Currently, over-fertilization is another burning environmental problem in many cropping regions of China. Finally, from a hydrological perspective, it also sounds not logical to minimize the water footprint in humid regions (e.g. in Southern China) where water does not limit wheat yields and where water scarcity is not a problem for the society nor the environment. What I'm questioning here is the one dimensional focus on water productivity in the current study which makes it impossible to draw useful conclusions from the results.

2.) The authors use a crop model to calculate water footprints but they completely miss to describe the model, its parametrization and its calibration. Therefore the results are not reproducible by external scientists. For example, the authors compare water footprints for warm and cold years. Temperature affects many different processes and to interpret the results of the study it is essential to know which effects have been considered in the model used here. How much is the difference in simulated evapotranspiration between cold and warm years and are the changes mainly an effect of different temperature or of associated differences in other variables, e.g. radiation or humidity? How is the effect of different temperature on crop yields? In general, higher temperature results in faster crop development and shortening of the period between sowing and maturity and therefore, in most cases, in lower yield. However, this effect can only be reflected in the model when the harvest date is considered dynamic. As far as I know AquaCrop offers two options: simulation with fix sowing and harvest dates and simulation with fix temperature sums. The shortening of the growing period can only be simulated when the second option is used. Therefore, description of the model parameterization and associated assumptions is essential. Furthermore, there is no information whether the model was calibrated, for which target variables the calibration was performed and which parameters were adjusted in the calibration process.

3.) The target variable for the study is the consumptive water footprint which requires simulation of evapotranspiration and crop yield. Previous research (e.g. all these recent model inter-comparison studies) indicated a high uncertainty in present model results for both variables. Since only one specific crop model has been used for the present study it is a challenge to prove the reliability of the results, in particular when considering that the reported differences shown between cold and warm years, irrigated and rainfed wheat, humid and arid regions are relatively low (Tables 2-6). How did the authors validate their results? The comparison of simulated yield to yields simulated with a another model (Figure 8) and the comparison of province level yields reported for one specific year with potential yield simulated by the authors provide little evidence that spatial patterns and temporal dynamics in water footprints simulated for this study are reliable. Therefore, the section describing the model validation needs to be extended and improved.

Specific comments: - Which process explains differences in the water footprint across soil classes for the irrigated winter wheat? If drought is the only stress factor considered in the study, the soil class should not have an effect for the irrigated winter wheat. - Tables 2-6: Do you really show averages in the last column or is it the median (50th percentile)?

---

## Author Comment (AC1) · 27 May 2016

**Authors' response to Interactive comments on "Benchmark levels for the consumptive water footprint of crop production for different environmental conditions: a case study for winter wheat in China"**

La Zhuo, Mesfin M. Mekonnen, Arjen Y. Hoekstra

l.zhuo@utwente.nl ; zhuo.l@hotmail.com

Dear Referee #1,

Thank you very much for your valuable comments and suggestions on our manuscript. We have provided our responses directly below the comments.

Anonymous Referee #1

The study presents an attempt to estimate benchmark levels for the consumptive water footprint of winter wheat in China by applying the crop model AquaCrop. Water footprints simulated for the period 1961-2008 are compared between dry and wet years, between warm and cold years, across soil types and between irrigated and rainfed wheat. Such an analysis is interesting in general and fits also well to the scope of the journal. However, I think that the manuscript requires substantial improvement before it may be considered for publication in HESS. May main points of criticism are:

1.) I completely miss a discussion on the relevance of the water footprints obtained in the present study. Why is this information needed? Farmers, for example, are not interested in optimizing water footprints; they are more interested in optimizing their economic return. The attempt to optimize the water footprint to benchmark levels provided in this study may also be misleading from an environmental or ecological perspective because these water footprints can only be achieved when nutrients are not limiting, thus require high nutrient inputs and consequently high nutrient losses. Currently, over-fertilization is another burning environmental problem in many cropping regions of China. Finally, from a hydrological perspective, it also sounds not logical to minimize the water footprint in humid regions (e.g. in Southern China) where water does not limit wheat yields and where water scarcity is not a problem for the society nor the environment. What I'm questioning here is the one dimensional focus on water productivity in the current study which makes it impossible to draw useful conclusions from the results.

**Response:** The purpose of developing water footprint (WF) benchmarks of a product is to provide an incentive for producers to reduce the WF of their products toward reasonable levels and thus use water as efficiently as possible (Hoekstra, 2013; 2014). In crop production, we agree with Referee #1 that farmers are more interested in optimizing their economic return. As water is one of the most fundamental resources for crop production and becoming increasingly limited for agriculture (Huang et al., 2002), undoubtedly, how to minimize the investment in water resources as well as water resources management and to maximum the production output at the same time (i.e. minimize the WF) are important issues in not only individual farmers' but also water governors' consideration for reaching their highest economic return. Therefore, setting WF benchmarks in crop production is necessary and essential for both water users and managers.

The magnitude of consumptive (green and blue) WF per tonne of a crop is determined by ET over the cropping period and crop yield. Therefore, reducing consumptive WF can be achieved by reducing ET or increasing crop yield. Mekonnen and Hoekstra (2014) summarized technology and practices to reduce the WF in crop production, which include three aspects: (i) increasing yield, (ii) reducing non-beneficial ET and (iii) enhancing effective use of rainfall. Evidently, it does not have to cost higher nutrient to increasing crop yield. Other than the simplest and direct way of increasing fertilization, wise soil and water management (e.g. appropriate tillage) and high technology improvement (e.g. breeding technology, plant biotechnology technology to improving crops' adaptation to natural stresses and diseases) (Huang et al., 2002) can also contribute to increasing crop yield.

The current results show that there is high potential to decrease consumptive WF of winter wheat in South China. South China has 81% of national blue water resources (Jiang et al., 2015). However, the risk of water shortage is increasing in the wet South with the operation of the South-to-North Water Transfer Project and the increasingly competition on the water resources by different sectors (Barnnet et al., 2015). Therefore, water saving and benchmarking WF for the South China are as equally important as for the drier North.

The current study, as an explorative study by taking winter wheat in China as the study case, aims to explore which environmental factors should be distinguished when determining benchmark levels for the consumptive WF of crops. We believe that the reported conclusions can serve as reference information for water managers and water scholars when estimating and setting WF benchmarks of crop production.

2.) The authors use a crop model to calculate water footprints but they completely miss to describe the model, its parametrization and its calibration. Therefore the results are not reproducible by external scientists. For example, the authors compare water footprints for warm and cold years. Temperature affects many different processes and to interpret the results of the study it is essential to know which effects have been considered in the model used here. How much is the difference in simulated evapotranspiration between cold and warm years and are the changes mainly an effect of different temperature or of associated differences in other variables, e.g. radiation or humidity? How is the effect of different temperature on crop yields? In general, higher temperature results in faster crop development and shortening of the period between sowing and maturity and therefore, in most cases, in lower yield. However, this effect can only be reflected in the model when the harvest date is considered dynamic. As far as I know AquaCrop offers two options: simulation with fix sowing and harvest dates and simulation with fix temperature sums. The shortening of the growing period can only be simulated when the second option is used. Therefore, description of the model parameterization and associated assumptions is essential. Furthermore, there is no information whether the model was calibrated, for which target variables the calibration was performed and which parameters were adjusted in the calibration process.

**Response:** We will add detailed content on model description, parameterization and calibration of the AquaCrop in the revised paper. In addition, we will discuss uncertainties from underling assumptions when modelling in the revised manuscript.

The core of methodology in the current study is to explore to which environmental factor the level of consumptive WF benchmark of crops most sensitive by making use of crop water productivity modelling. Therefore, we did not calibrate the simulated crop yield according to real statistics. We calibrated input crop parameters including crop calendar, reference harvest index, and maximum root depth for China's winter wheat. We validated the simulated WFs of winter wheat by comparing to available database with similar assumptions under similar hypothetical conditions, as shown in Section 3.7.

In order to be more clear, as suggested by Referee #1, we will add all the detailed information on the parameter calibration and improve the content on result validation in the revised manuscript.

3.) The target variable for the study is the consumptive water footprint which requires simulation of evapotranspiration and crop yield. Previous research (e.g. all these recent model inter-comparison studies) indicated a high uncertainty in present model results for both variables. Since only one specific crop model has been used for the present study it is a challenge to prove the reliability of the results, in particular when considering that the reported differences shown between cold and warm years, irrigated and rainfed wheat, humid and arid regions are relatively low (Tables 2-6). How did the authors validate their results? The comparison of simulated yield to yields simulated with a another model (Figure 8) and the comparison of province level yields reported for one specific year with potential yield simulated by the authors provide little evidence that spatial patterns and temporal dynamics in water footprints simulated for this study are reliable. Therefore, the section describing the model validation needs to be extended and improved.

**Response:** The current study goal is exploring which *environmental factors* should be distinguished when determining consumptive WF of crop production. As an explorative study, we make use of modelling results under a kind of hypothetical condition by only considering water stress impacts in crop growth, which is aiming to avoid effects of non-environmental factors (e.g. technology, fertilization). Therefore, it is not possible to calibrate or validate the result according to real statistics. What the best we can do and we have done is comparing the current simulated crop yields to available few data based on similar hypothetical modelling or simulations, as we described in the Section 3.7 Discussion. In addition, the performance of AquaCrop on crop water use and yield simulation has been widely tested and evaluated for variety of crop types under different conditions (e.g. Kumar et al., 2014; Jin et al., 2014; Katerji et al., 2013; Abedinpour et al., 2012; Mkhabela and Bullock, 2012; Andarzian et al., 2011; Stricevic et al., 2011; Heng et al., 2009; Farahani et al., 2009; García-vila et al., 2009). Furthermore, the core result reported in the current study is the relative differences between the consumptive WFs for different level of certain input variables that diminishes the uncertainties in the absolute magnitude of simulated results on crop ET, yield and finally consumptive WF. Therefore, we believe that the current reported results are valid and comparable to the results carried out by different crop models and can serve as useful referential information for water managers when setting WF benchmarks.

As suggested, we will improve the content on model and result validation by summarizing available relative information on the model performance in previous studies in the revised manuscript.

Specific comments:

- Which process explains differences in the water footprint across soil classes for the irrigated winter wheat? If drought is the only stress factor considered in the study, the soil class should not have an effect for the irrigated winter wheat.

**Response:** For the irrigated winter wheat, the different levels of ET defined the differences in the consumptive WF across soil classes. As we interpreted in the line 20-23 in page 7, the WF benchmarks for irrigated winter wheat in sandy soils are about 15% smaller than the WF benchmarks for the other three soil classes, due to relatively low ET. The low ET of sandy soil was resulted from the fast percolation of water below the root zoon.

-Tables 2-6: Do you really show averages in the last column or is it the median (50[th] percentile)?

**Response:** We show the weighted averages in the last column in Tables 2-6.

**References:**

Abedinpour, M., Sarangi, A., Rajput, T.B.S., Singh, M., Pathak, H. and Ahmad, T.: Performance evaluation of AquaCrop model for maize crop in a semi-arid environment, Agricultural Water Management, 110, 55-66, doi: 10.1016/j.agwat.2012.04.001, 2012.

Andarzian, B., Bannayan, M., Steduto, P., Mazraeh, H., Barati, M.E., Barati, M.A. and Rahnama, A.: Validation and testing of the AquaCrop model under full and deficit irrigated wheat production in Iran, Agricultural Water Management, 100(1), 1-8, doi: 10.1016/j.agwat.2011.08.023, 2011.

Barnett, J., Rogers, S., Webber, M., Finlayson, B. and Wang, M.: Sustainability: Transfer project cannot meet China's water needs, Nature, 527 (7578), 295-297, doi: 10.1038/527295a, 2015.

Farahani, H., Izzi, G. and Oweis, T.Y.: Parameterization and evaluation of the AquaCrop model for full and deficit irrigated cotton, Agronomy Journal, 101(3), 469-476, doi: 10.2134/agronj2008.0182s, 2009.

García-vila, M., Fereres, E., Mateos, L., Orgaz, F. and Steduto, P.: Deficit irrigation optimization of cotton with AquaCrop, Agronomy Journal, 101(3), 477-487, doi:10.2134/agronj2008.0179s, 2009.

Heng, L.K., Hsiao, T.C., Evett, S., Howell, T. and Steduto, P.: Validating the FAO AquaCrop model for irrigated and water deficient field maize, Agronomy Journal, 101(3), 488-498, doi: 10.2134/agronj2008.0029xs, 2009.

Hoekstra, A. Y.: The water footprint of modern consumer society, Routledge, London, UK, 2013.

Hoekstra, A. Y.: Sustainable, efficient, and equitable water use: the three pillars under wise freshwater allocation, Wiley Interdisciplinary Reviews: Water, 1, 31-40, doi: 10.1002/wat2.1000, 2014.

Huang, J., Pray, C. and Rozelle, S.: Enhancing the crops to feed the poor, Nature, 418 (6898), 678-684, doi:10.1038/nature01015, 2002.

Jiang, Y.: China's water security: Current status, emerging challenges and future prospects, Environmental Science & Policy, 54, 106-125, doi:10.1016/j.envsci.2015.06.006, 2015.

Jin, X. L., Feng, H. K., Zhu, X. K., Li, Z. H., Song-S. N., Song, X. Y., Yang, G. J., Xu, X. G. and Guo, W. S.: Assessment of the AquaCrop model for use in simulation of irrigated winter wheat canopy cover, biomass, and grain yield in the North China Plain, PLoS ONE, 9(1), e86938, doi: 10.1371/journal.pone.0086938, 2014.

Kumar, P., Sarangi, A., Singh, D.K. and Parihar, S.S.: Evaluation of AquaCrop model in predicting wheat yield and water productivity under irrigated saline regimes, Irrigation and Drainage, 63(4), 474-487, doi: 10.1002/ird.1841, 2014.

Mekonnen, M. M., and Hoekstra, A. Y.: Water footprint benchmarks for crop production: A first global assessment, Ecol Indic, 46, 214-223, doi: 10.1016/j.ecolind.2014.06.013, 2014.

Mkhabela, M.S. and Bullock, P.R.: Performance of the FAO AquaCrop model for wheat grain yield and soil moisture simulation in Western Canada, Agricultural Water Management, 110, 16-24, doi: 10.1016/j.agwat.2012.03.009, 2012.

Stricevic, R., Cosic, M., Djurovic, N., Pejic, B. and Maksimovic, L.: Assessment of the FAO AquaCrop model in the simulation of rainfed and supplementally irrigated maize, sugar beet and sunflower, Agricultural Water Management, 98(10),1615-1621, doi: 10.1016/j.agwat.2011.05.011, 2011.

---

## Referee Comment (RC2) · Anonymous Referee #2 · 4 Jun 2016

The paper presents a study for determining benchmark levels for the consumptive water footprint (WF of winter wheat production in China, considering the influence of different external environmental factors (rainfed vs irrigated, wet vs dry years, warm vs cold years, and across soil classes and climate zones). This is done using the FAO's crop water productivity model AquaCrop. Only water stress is considered as limiting factor. The objective seems to be to identify where WF reductions should be targeted. China is the main wheat producer in the world an increasing water use efficiency in wheat production is certainly a priority, especially for water stress regions. In this sense, the topic of study is certainly interesting and appealing and within the scope of the journal. However, I have several important concerns on the research presented:

1) Green and blue water inputs are not separated for irrigated agriculture, while this has been done in previous studies (eg Liu et al. (2009), Global consumptive water use for crop production: The importance of green water and virtual water, Water Resour. Res., 45, W05428). Please justify why this separation is not considered in this study. This distinction seems fundamental for the practical relevance of the study, since the WF reduction will mainly have implications in blue water, correct?

2) Besides the previous mentioned limitation, I miss more insight on the practical interest of the results. In order to be able to defiне where WF improvements are possible and what measures to take to create higher levels of water productivity, the factors that determine the current levels of water productivity must be understood. However, this approach does not really help much on this sense. Climate cannot be controlled and the influence of the managerial factors are not incorporated in this study. So for me it is unclear to what extend the differences found in WF values are just due to the local conditions and cannot be significantly modified.

3) The description on the modelling assumptions and calibration is too limited and it is needed for properly understanding the simulation done.

---

## Author Comment (AC2) · 10 Jun 2016

**Authors' response to Interactive comments on "Benchmark levels for the consumptive water footprint of crop production for different environmental conditions: a case study for winter wheat in China"**

La Zhuo, Mesfin M. Mekonnen, Arjen Y. Hoekstra

l.zhuo@utwente.nl ; zhuo.l@hotmail.com

Dear Referee #2,

We appreciate very much for your valuable comments and suggestions on our manuscript. We have provided our responses directly below the comments.

Anonymous Referee #2

The paper presents a study for determining benchmark levels for the consumptive water footprint (WF of winter wheat production in China, considering the influence of different external environmental factors (rainfed vs irrigated, wet vs dry years, warm vs cold years, and across soil classes and climate zones). This is done using the FAO's crop water productivity model AquaCrop. Only water stress is considered as limiting factor. The objective seems to be to identify where WF reductions should be targeted. China is the main wheat producer in the world an increasing water use efficiency in wheat production is certainly a priority, especially for water stress regions. In this sense, the topic of study is certainly interesting and appealing and within the scope of the journal. However, I have several important concerns on the research presented:

1) Green and blue water inputs are not separated for irrigated agriculture, while this has been done in previous studies (eg Liu et al. (2009), Global consumptive water use for crop production: The importance of green water and virtual water, Water Resour. Res., 45, W05428). Please justify why this separation is not considered in this study. This distinction seems fundamental for the practical relevance of the study, since the WF reduction will mainly have implications in blue water, correct?

**Response:** We agree with Referee #2 that the separation of green and blue water consumption is important for irrigated agriculture (Liu et al., 2009). The green and blue water footprint (WF, $m^3 t^{-1}$) of a crop within a grid cell is calculated as the actual green and blue evapotranspiration (ET, $m^3 ha^{-1}$) over the growing period divided by the crop yield (Y, t $ha^{-1}$), separately. We separated the green (precipitation) and blue water (irrigation) inputs as well as the resulted

green and blue WF over the cropping period through the AquaCrop modelling as following Chukalla et al. (2016) and Zhuo et al. (2016). The separation of green and blue ET, was carried out by tracking the daily green and blue soil water balances based on the contribution of rainfall and irrigation, respectively. As suggested also in the comment (3), we will add the detailed description on AquaCrop modelling and assumptions in the revised manuscript.

In the current study, we did not distinguish green and blue WF benchmarks with two reasons. Firstly, the ratio of green to blue WF of a crop heavily depends on local green water resources availability, which is defined by the climate of certain time in a certain location. Location-specific blue WF benchmarks can be developed as a function of the overall consumptive WF benchmarks and local green water availability (Mekonnen and Hoekstra, 2014). Secondly, the purpose of the current study is to find out to which environmental factor the consumptive WF benchmark is most sensitive. The conclusion for agricultural water management at a large spatial scale only can be done by looking at the combined green-blue WF benchmarks given the first reason. However, as we mentioned in the last paragraph in 3.7 Discussion that, for each specific location, the blue WF benchmark which is translated from a certain benchmark of the consumptive WF of a crop is curtail for each specific location as a function of the local green water availability.

Regarding the implications for reducing consumptive WF of irrigated crops, not only the reduction of blue WF (e.g. improved irrigation technology), but also yield increase (e.g. soil nutrient management, weed control, crop variety selection) and reduction of green WF (e.g. crop scheduling, mulching) play the main roles (Hoekstra, 2013; Mekonnen and Hoekstra, 2014).

2) Besides the previous mentioned limitation, I miss more insight on the practical interest of the results. In order to be able to deïnˇA¸ne where WF improvements are possible and what measures to take to create higher levels of water productivity, the factors that determine the current levels of water productivity must be understood. However, this approach does not really help much on this sense. Climate cannot be controlled and the influence of the managerial factors are not incorporated in this study. So for me it is unclear to what extend the differences found in WF values are just due to the local conditions and cannot be significantly modified.

**Response:** Crop growth and water use are driven by environmental conditions that cannot be controlled by humans (e.g. climate, soil texture) and managerial factors (e.g. fertilizer,

irrigation, tillage, mulching) (Zwart et al., 2010; Brauman et al., 2013). Apparently, understanding the sensitivity of the WF reduction potential or benchmarks of growing a crop to the uncontrollable environmental factors is crucial for both managers and farmers to better and effectively modify the controllable factors to improve the crop water productivity and to reduce the consumptive WFs in practice.

The current study investigates which *environmental factors* (climate, soil) should be distinguished when determining consumptive (green and blue) WF *benchmarks* of crop production, by taking winter wheat in China as the study case. In the results, the 26-31% smaller WF benchmarks for the humid zone than for the arid zone indicates that there are significant different levels of the limit or potential of reducing consumptive WF of winter wheat for different climate zones. For water managers, it is an important information when setting WF benchmarks of a crop for a region including different climate zones. Meanwhile, such information is also fundamental for wise water allocation and fair share of water resources among different sectors for a region (Hoekstra, 2013).

We will add the sentences stating the practical significance of the current study in the Introduction of the revised manuscript.

3) The description on the modelling assumptions and calibration is too limited and it is needed for properly understanding the simulation done.

**Response:** Yes, as suggested, we will add the detailed description on the AquaCrop modelling process, assumptions and calibration in the revised manuscript.

**References:**

Chukalla, A., Krol, M., and Hoekstra, A.: Green and blue water footprint reduction in irrigated agriculture: effect of irrigation techniques, irrigation strategies and mulching, Hydrol Earth Syst Sc, 19, 4877-4891, doi: 10.5194/hess-19-4877-2015, 2015.

Brauman, K. A., Siebert, S., and Foley, J. A.: Improvements in crop water productivity increase water sustainability and food security—a global analysis, Environ Res Lett, 8, 024030, doi:10.1088/1748-9326/8/2/024030, 2013.

Hoekstra, A. Y.: The water footprint of modern consumer society, Routledge, London, UK, 2013.

Hoekstra, A. Y.: Sustainable, efficient, and equitable water use: the three pillars under wise freshwater allocation, Wiley Interdisciplinary Reviews: Water, 1, 31-40, doi: 10.1002/wat2.1000, 2014.

Liu, J., Zehnder, A. J. B., and Yang, H.: Global consumptive water use for crop production: The importance of green water and virtual water, Water Resour. Res., 45, W05428, doi:10.1029/2007WR006051, 2009.

Mekonnen, M. M., and Hoekstra, A. Y.: Water footprint benchmarks for crop production: A first global assessment, Ecol Indic, 46, 214-223, doi: 10.1016/j.ecolind.2014.06.013, 2014.

Zhuo, L., Mekonnen, M. M., Hoekstra, A. Y., and Wada, Y.: Inter- and intra-annual variation of water footprint of crops and blue water scarcity in the Yellow River basin (1961–2009), Advances in Water Resources, 87, 29-41, doi: 10.1016/j.advwatres.2015.11.002, 2016.

Zwart, S. J., Bastiaanssen, W. G. M., de Fraiture, C., and Molden, D. J.: A global benchmark map of water productivity for rainfed and irrigated wheat, Agricultural Water Management, 97, 1617-1627, doi: 10.1016/j.agwat.2010.05.018, 2010.

---

## Author Response (AR1)

**Authors' Response**

Dear Prof. Harrie-Jan Hendricks Franssen,

Thank you for your suggestions and comments on our manuscript entitled "Benchmark levels for the consumptive water footprint of crop production for different environmental conditions: a case study for winter wheat in China" (ID: hess-2016-47). We appreciate the opportunity to revise the paper. Based on the suggested amendments and comments from the Editor and two Referees, we carefully revised the manuscript. The revised parts are in RED color in the updated manuscript.

In response to Referee#1 comment (1) and Referee#2 comment (2), in Section 1 Introduction, we elaborate on the practical relevance of the current study. When water is scarce, raising production per unit of water (i.e. increasing water productivity in terms of t $m^{-3}$ or reducing the WF in $m^3 t^{-1}$) is a key challenge in order to save water and achieve sustainable water use at catchment level. We will need WF benchmarks as reference of what water use per unit of crop is reasonable. The current paper explores what variables need to be considered when setting WF benchmarks (and finds that climate is the most important factor when setting WF benchmarks). In Section 3.6 we add a reflection on the significance of saving water and WF benchmarking in the wet South China.

In response to Referee #2 comment (1), we add the information in Section 2.2 explaining why we did not distinguish green and blue WF benchmarks in the current study. We do, however, of course, distinguish between green and blue ET in the simulations, as elaborated now in more detail in Section 2.

As suggested by both Referees, in the revised manuscript, we split Section 2 Method and Data into two subparts: 2.1 Estimating consumptive WF of growing a crop and 2.2 Benchmarking consumptive WF of growing a crop. We added detailed information on the AquaCrop model, parameterization, calibration and assumptions in Section 2.1 as suggested by both Referees. In 3.7 Discussion, we add sentences on results validation and the effects on core results and conclusions from the uncertainties in absolute magnitude of simulated WFs.

In response to Referee#1 comment (3) we add in Section 3.7 Discussion a discussion of the need for using different models to identify uncertainties around the WF benchmark values presented.

We thank again all the efforts of Editor and two Referees on our manuscript.

Sincerely,

[revised manuscript text omitted]

---

## Author Response (AR2)

**Authors' Response**

Dear Prof. Harrie-Jan Hendricks Franssen,

Thank you very much for your final decision that our manuscript entitled "Benchmark levels for the consumptive water footprint of crop production for different environmental conditions: a case study for winter wheat in China" is accepted to be published in HESS. We are very happy!

We have corrected the three points as you commented in the revised manuscript. The revised places are colored in RED in the updated text. There are some changes happened regarding all of three authors' affiliations, we also update the related information.

We thank again all the efforts of Editor and two Referees on our manuscript.

Sincerely,

La Zhuo, Mesfin M. Mekonnen, Arjen Y. Hoekstra